# Revealing Consequences of the Husking Process on Nutritional Profiles of Two Sorghum Races on the Male Sterility Line

**DOI:** 10.3390/foods13071100

**Published:** 2024-04-03

**Authors:** Maha Khalfalla, László Zsombik, Zoltán Győri

**Affiliations:** 1Faculty of Agricultural and Food Sciences and Environmental Management, Institute of Nutrition, University of Debrecen, Böszörményi út 138, 4032 Debrecen, Hungary; gyori.zoltan@unideb.hu; 2Central Laboratory, Ministry of Higher Education and Scientific Research, Khartoum P.O. Box 7099, Sudan; 3Research and Educational Farm, University of Debrecen, Vilmos út 4–6, 4400 Nyíregyháza, Hungary; zsombik@agr.unideb.hu

**Keywords:** mineral contents, total protein, husking process, color profile, *S. bicolor*, Kafirin

## Abstract

The male sterility line is a vital approach in the genetic breeding of sorghum. The husking process affects the grain’s nutritional composition, emphasizing the intricate relationship between genetic enhancement and dietary requirements. The current study assessed the influence of the Husking Fraction Time Unit (HFTU) process, which was set at 30 (S) and 80 (S) time units per second (S). The study assessed the impact of the (HFTU) process on fifty-one inbred line sorghum race varieties, which implied diverse nutritional profiles considering the pericarp color variations. The assessment of the nutritional profile involved dry matter, total protein, and minerals (P, K, S, Ca, Mg, Na, Fe, Zn, and Mn). The variety groups showed a significance value of *p* ≤ 0.05, indicating the study hypothesis’s truth. While results demonstrated substantial impacts implied by the Husking Fraction Time Unit (HFTU) technique, the occurrence was noted when the dry matter percentage was increased in the husked products, specifically the endosperm (grits) and bran. Conversely, the protein variation percentage between the bran and endosperm (grits) for the *S. bicolor* race was calculated at 33.7%. In comparison, the percentage was 11.8% for the Kafirin race. The 80 (S) time unit, on the other hand, had an observable effect on the mineral reconcentration when the Kafirin race had the highest averages of K mg/kg^−1^, Ca mg/kg^−1^, and Fe mg/kg^−1^, which were 5700.5 mg/kg^−1^, 551.5 mg/kg^−1^ and 66.5 mg/kg^−1^, respectively. The results of this study could benefit breeders and nutrition specialists in developing genotypes and processing sorghum grains, promoting research, and aiding several industrial sectors owing to the grain’s adaptability and nutritional properties.

## 1. Introduction

Sorghum (*Sorghum bicolor* L.) is a productive crop worldwide, especially in dry locations, and it plays a significant role in supplying essential sustenance to millions of people, particularly in developing nations [1]. Although sorghum processing is important, it requires additional research for industrialized countries compared to renowned crops such as maize, wheat, and rice [2]. *S. bicolor* is recognized as a staple drought-tolerant crop worldwide. In response to climate change and its contribution to reducing CO_2_ emissions, sorghum has been increasingly cultivated in various regions abroad [3,4]. Sorghum is well recognized as a lucrative cereal grain crop due to its high productivity and ability to thrive in many conditions [5,6]. Climate change has increased Mediterranean sorghum production and stimulated research in food sciences [7,8,9]. Therefore, the advancement of sorghum grain processing has resulted in its heightened utilization due to its nutritional attributes and advantages for those with gluten intolerance [10,11]. For example, Kafirins, known as the main storage proteins in sorghum, significantly influence its qualitative properties, such as their appropriateness for food, feedstock, and biomaterial uses, which makes them an essential factor in sorghum quality and improvement measurements [12]. The husking process for sorghum grain involves removing the outer layer to reach the edible kernel, which could improve taste and digestion [2]. Hence, grain processing is a viable approach to enhance the bioavailability of minerals in processed sorghum grain [13]. Processed products such as bran are the richest anatomical parts with mineral contents and carbohydrates, and they can fit various food applications [13]. Processing and promoting its industrial applications can effectively control sorghum’s anti-nutritional content and functional qualities [14]. All the mentioned characteristics make sorghum a useful dietary choice with promising prospects in food security [15].

Male sterile lines in sorghum grains are essential for the generation of hybrids and the enhancement of grains. Researchers exploit heterosis to enhance production and crop characteristics [16,17], exploring male sterility induction techniques in sorghum and focusing on several ways to improve yield and quality in sorghum breeding programs [18,19]. Developers use controlled hybridizations, male sterility lines, crop growth patterns, and pollination tendencies in sorghum breeding to optimize nutritional qualities [20]. Despite the valuable insight about sorghum in male sterility lines, the relationship between the husking process and the nutritional characteristics of sorghum, especially in male sterility lines, needs to be better recognized. For several reasons, it is crucial to comprehend the effects of husking on the nutritional content of diverse sorghum, especially on male sterility lines. Firstly, it guarantees that sorghum grains are used to their fullest potential in human and animal feeding, maximizing their nutritional advantages [21]. Understanding the relationship between male sterility lines in sorghum breeding programs and grain processing procedures is crucial for sustainable agricultural growth due to the increasing significance of male sterility line enhancements [22,23]. The inbred line varieties can serve as the genetic basis for breeding programs to enhance crops’ nutritional value [24]. Specifically, the previous investigations indicated the efficiency of time-unit processing on sorghum nutritional properties [25]. Furthermore, sufficient grain processing could provide significant information for food processing firms to create effective food processing that reduces nutrient loss [24].

On the other hand, the grain color profile is crucial since it directly impacts the grain’s nutritional composition [20]. Sorghum exhibits variations in pericarp color, involving white, black, and red, which might affect the nutritional and antioxidant characteristics due to the flavonoid concentration in the aleurone layer and seed coat of the grain [26]. The color of the grain is mostly governed by genetic regulation, with several alleles controlling the grain’s color [24]. Sorghum color characteristic measurements are important due to their capability to gain insight into its nutritional content, processing compatibility, and consumer appeal, affecting its use in food, feed, and industrial purposes [27,28]. The Konica Minolta CR-410 colorimeter is used for general color categorization [29,30], as its outcomes were detailed as relevant findings.

This research aimed to determine how the husking process over 30 (S) and 80 (S) time units affected the nutritional value of two races of sorghum grown on male sterility lines. The present study measured the impacts on the major nutrient contents such as dry matter, proteins, and minerals (P, K, S, Ca, Mg, Cu, Fe, Zn, Na, and Mn) in sorghum grains as an attempt to determine and measure how the overall nutritional quality of the grains had improved. Additionally, by comparing these effects across various sorghum races, one might aim to identify race-specific differences that can impact nutritional results, contributing to the diverse races’ future improvements.

## 2. Material and Methods

### 2.1. Instruments and Reagents

#### 2.1.1. Crude Protein Measurements

We purchased the distillation unit system from VWR International Ltd., Radnor, PA, USA.

The sulfuric acid (*c*(H_2_SO_4_) = 18 mol/L, *ρ*20(H_2_SO_4_) = 1.84 g/mL), the boric acid (aqueous solution, [*ρ*20(H_3_BO_3_) = 40 g/L]), and the catalyst (CuSO_4_·5H_2_O) were all bought in Hungary from different lab supply retailers. The items varied depending on where they originated (Vwr International Kft, Debrecen, Hungary and Thermo Fisher Scientific, Budapest, Hungary).

#### 2.1.2. Mineral Content Measurements

The system iCAPTM 7400, an ICP-OES analyzer, was sourced from PerkinElmer Inc., Waltham, MA, USA—catalogue number 84230074181.

Ultra-pure water was sourced from Millipore, S.A.S. (Molesheim, France), while the chemical reagents were sourced from VWR International Ltd. (Geldenaaksebaan, Belgium).

In accordance with [31], we used BCR CRM 189 (whole grain) for quality accreditation from the International Plant Exchange Network (University of Wageningen, The Netherlands).

#### 2.1.3. The Husker

TM05C husker equipment was sourced from Satake Engineering Co. (Hiroshima, Japan).

#### 2.1.4. Colorimeter Camera

Konica Minolta, Chroma Metre (CR-410, 2002), originated from Minolta Co., Ltd., (Tokyo, Japan).

### 2.2. Material and Sample Preparations

We acquired identical samples of two sorghum racial varieties from Alpha Seed Breeding House in Karcag. These samples were categorized into two races (fifty-one inbred lines), *S. bicolor* (twenty-one) and Kafirin (thirty). The samples were collected at various ripening times, with S. bicolor classified into early and middle ripening times. In contrast, the Kafirin races were documented as a late ripping stage. The experiment on male sterility included three lines: a sterility line and a restorer line, while the Alpha 12 variety was performed on a maintainer line. The samples were dried, and the grain samples were finely processed using the Retsch SK-3 hammer mill with a 1 mm sieve. The careful grinding process ensured that the samples were homogenized. The milled whole grains were compared to the dehulled grains for the investigation outcomes.

### 2.3. Application of Husking Procedure

We employed the TM05C husker [32] for the husking process. A total of 50 g of samples were weighted into a husker with a 46–60 mesh abrasive wheel and 54 Kw power. The abrasive wheel rotated at 800–1100 rpm and was sifted through a No. 60 mesh sieve (4760 µm). Diverse grain size diameters were tested, ranging from 3.0 mm, 3.2 mm, 3.6 mm, 4.0 mm, to 4.5 mm, and the grain size of 4.0–4.5 mm was the most sufficient and was selected for the Husking Fraction Time Unit (HFTU) procedure measurements.

### 2.4. Determination of the Crude Protein

The crude protein contents were analyzed using the Kjeldahl procedure [33]. The tube was inserted into a block heater set at 420–430 °C for 2 h. Following digestion, the samples were left to cool. Furthermore, we used Converter 6.25 to determine the total protein content.

### 2.5. Determination of Dry Matter

The following steps measured the dry matter content.

Drying Procedure: The sample was put in an oven set to a temperature range of 130–135 °C. The sample was dried in the oven until it reached a consistent weight. All moisture content was successfully eliminated from the model.

Measuring the weight of the dehydrated sample: The desiccated sample was cautiously removed from the oven. The dried sample’s weight was documented [34]. The sample’s dry matter content was determined using the following formula:(1)$$Dry\;Matter\;Content\left(\%\right)=\frac{Weight\;of\;Dried\;Sample}{Weight\;of\;Original\;Sample}\times 100$$

### 2.6. Determination of Mineral Elemental Contents

The mineral contents of the grain samples were analyzed at the Central Chemical Laboratory of the Agricultural Centre, University of Debrecen. We validated the measurements using a genuine wheat sample, BCR CRM 189 (whole grain), for quality described by [31]; the measurements were carried out in many stages [35]. We used the iCAP 7400 and inductively coupled plasma optical emission spectroscopy (ICP-OES) to measure elements at specific wavelengths: P 177.495 nm, K 404.721 nm, S 183.801 nm, Ca 183.034 nm, Mg 285.204 nm, Na 330.237 nm, Cu 324.754 nm, Fe 238.204 nm, Zn 213.856 nm, and Mn 259.373 nm. We measured 1 g of each sample. The materials underwent aqueous acid digestion, including predigesting and digestion stages. The samples were subjected to heat at 60 °C for 30 min using model block digestion equipment (MIM OE-718/A) after adding 10 mL of nitric acid (HNO_3_, 69% *v/v*). After a brief cooling period, 3 cm^3^ of hydrogen peroxide (H_2_O_2_, 30% *v/v*) was introduced to the samples, which were then moved to the first digestion phase. We raised the temperature of the digestor to 120 °C and maintained it for 90 min before switching it off and letting it stabilize for 10 to 20 min. The capacity was raised to 50 cm^3^ using ultra-pure water. We filtered the homogenized suspension using an MN 640W filter paper. The elements’ concentration was indicated based on the weight of the grains when dry, measured in mg/kg^−1^.

### 2.7. Evaluation of Husked Grain Color

The husked grain colors were measured in terms of L* (whiteness), a* (redness), b* (yellowness), and Y (brightness) values using the Konica Minolta Camera (Chroma meter, CR-410, Minolta Co., Ltd., Tokyo, Japan, 2002).

### 2.8. Statistical Analysis

The statistical analysis was performed using SPSS 28.0 software. Descriptive data were used to show the sample size of the original data findings for all variables. The data sets were assessed for Gaussian normality, confirming normal distributions in dry matter, total protein, and mineral contents. An ANOVA one-way analysis was used to determine the variation in dry matter and total protein. Additionally, an ANOVA one-way model was used to analyses differences in mineral content independently. A linear regression was performed for the color profile analysis, and Pearson correlation was used to determine statistical significance at a significance level of *p* ≤ 0.05. The visualization was created using SAS 17 software.

## 3. Results

The results showed that the HFTU process based on 30 (S) and 80 (S) time units positively impacted the redistribution and reconcentration of the nutritional contents on the husked product (bran and endosperm (grits)) attributed to the two inbred line sorghum races, as explained in Table 1 and Table 2. When the impact of the HFTU process was highlighted, the changes in dry matter, total protein, P, K, S, Ca, Mg, Cu, Fe, Zn, Na, and Mn were attributed to two fractions of milling (husker) compared to whole grain (ground). The mechanism of fraction separation facilitated an understanding of the HFTU process’s impact on nutritional concentrations and accumulation in each fraction milling (bran and endosperm). This analysis was carried out based on the difference between the *S. bicolor* and Kafirin inbred-line race varieties. Correlations were performed for most associated mineral contents that can affect each other, such as Ca: P, K: Na, and Fe: Zn, as shown in Figure 1 and Figure 2.

### 3.1. Influence of the HFTU Process on the Dry Matter and Crude Protein

The findings showed that dry matter was influenced by the HFTU process, as evidenced by the reconcentration of the husked products compared to the ground whole grains (Table 1 and Table 2). At the same time, no significant variation was observed among the respective varieties (*p* < 0.05) in the case of the husked products of the *S. bicolor* and Kafirin races.

The efficiency evaluations of the HFTU process were evaluated according to protein % redistribution, reconcentrate, and accumulation in the husked product (bran and endosperm (grits)) attributed to the sorghum races with various ripping stages. These evaluations are detailed in Table 1 and Table 2 (*p* < 0.05), where the variation in the crude protein percentage was 19.2% based on the ground whole grains/endosperm (grits) and 10.4% based on the whole grains/bran. The highest protein content was observed within 1.7% in the endosperm (grits) compared to 0.87% in the bran. This was evidenced by 28.9% of the total varieties having a higher concentration of protein % in the endosperm compared to the bran attributed to the different ripping stages; (Alpha 4) the *S. bicolor* variety showed the highest protein % content in the endosperm within 1.5% and 1.2% in the bran.

### 3.2. Influence of the HFTU Process on the Mineral Contents

#### 3.2.1. Phosphorus

The ripping stages showed mentionable implications of the HFTU process on the P mg/kg^−1^ in the diverse investigated inbred *S. bicolor* varieties (Table 1 and Table 2), with a significance value of *p* > 0.05. The highest P mg/kg^−1^ levels were found in the bran, at 7985 mg/kg^−1^, compared to 3137 mg/kg^−1^ in the whole grains. The difference between the whole grains and the bran was 154.6%. In contrast, the lowest accumulation was observed in the endosperm grits attributed to early, middle, and late ripping stages, as shown in Figure 1 and Figure 2. At the same time, the variance between 30 (S) and 80 (S) time units of the bran was found to be within 18.1%.

#### 3.2.2. Potassium

Potassium mineral contents were abundant in the respective varieties, as shown in Table 1 and Table 2. The K mg/kg^−1^ contents varied among sorghum races, with the Alpha 4 *S. bicolor* variety (early ripping) exhibiting the highest average of 9754.7 mg/kg^−1^ attributed to 80 (S) time units, as shown in Figure 1 and Figure 2, particularly in the case of Kafirin races. K mg/kg^−1^ did not show a significance value (*p* > 0.05) in the case of the bran for the two tested sorghum races.

#### 3.2.3. Sulfur

The results showed that S mg/kg^−1^ contents were affected by the HFTU process. This was attributed to 30 (S) time units in the case of the Kafirin races, where average values of 1078 mg/kg^−1^ and 978.8 mg/kg^−1^, respectively, were demonstrated, with a significance value of *p* < 0.05. In addition, sulfur minerals showed a good ratio of N:S, ranging from 14:1 in *S. bicolor* to 12:1 in Kafirin samples.

#### 3.2.4. Calcium

The results of Ca mg/kg−1 content obtained from the Husking Fraction Time Unit (HFTU) procedure are presented in Figure 1 and Figure 2. In the case of *S. bicolor,* the Ca mg/kg^−1^ contents were recorded at 373.6 mg/kg^−1^ for whole grain, 268.9 mg/kg^−1^ for endosperm (grits), and 566.7 mg/kg^−1^ for bran. In contrast, the Kafirin race showed averages of 494.5 mg/kg^−1^, 381.2 mg/kg^−1^, and 716.3 mg/kg^−1^ in the case of the Ca mg/kg^−1^ in the endosperm (grits), with a significance value *p* < 0.05.

Table 1 and Table 2 show that the husking process and variations in ripping times influenced the Ca mg/kg^−1^ contents. We calculated the ratio of Ca:P, which can facilitate the estimation of the phytic acids in the tested races, as shown in Figure 1 and Figure 2. It is demonstrated at 1:15 in the *S. bicolor* race and 1:11 in the Kafirin race.

#### 3.2.5. Magnesium

The results showed that the HFTU process influenced the Mg mg/kg^−1^ contents of the varieties’ whole grain and dehulled grains based on the two sorghum races, as shown in Table 1 and Table 2. This is because the 30 (S) time units showed the highest Mg mg/kg^−1^ concentration within 1657.1 mg/kg^−1^ compared to 1053.4 mg/kg^−1^ attributed to the 80 (S) time units in Figure 1. Despite the mentionable variation in Mg mg/kg^−1^ among the respective sorghum races, the Alpha 51 variety showed a reduction in Mg mg/kg^−1^ compared to whole grain, endosperm (grits), and bran. Statistically significant differences were observed (*p* < 0.05) in the case of whole grains and bran. In contrast, the variation ratio of whole grain to bran was demonstrated at 5:1.

#### 3.2.6. Sodium

Sodium (Na) mineral content in the cereal grains is a crucial measurement due to the healthy diet issue; the HFTU process showed a significant impact on the Na mg/kg^−1^ redistribution and reconcentration on the husked *S. bicolor* grains (Table 1 and Table 2). The result showed a reduction in Na mg/kg^−1^ levels, as seen on the bran, compared to the endosperm (grits). The *S. bicolor* and Kafirin samples had average levels of 36.9 mg/kg^−1^ and 61.8 mg/kg^−1^, respectively, over 30 (S) time units. At the same time, 36.3 mg/kg^−1^ and 54.4 mg/kg^−1^ were attributed to 80 (S) time units, with a significance value of *p* < 0.05.

On the other hand, the observed ratio of K:Na was found to be within 97:1, compared to 69:1 in the case of ground (whole grains), as calculated across all sorghum race varieties, according to the results shown in Table 1 and Table 2.

#### 3.2.7. Copper

Copper (Cu) mg/kg^−1^ showed an acceptable range among the investigated mineral contents. The observed average in the husked products endosperm (grits) and bran ranged from 2.8 mg/kg^−1^ to 10 mg/kg^−1^ (Table 1 and Table 2); the bran recorded the highest average within 10.0 mg/kg^−1^. In contrast, the ratio of the bran/whole grain ratio was observed to be 3:1 for the *S. bicolor* races, with a significance value of *p* < 0.05, while the highest average of 16 mg/kg^−1^ was observed among Kafirin races (*p* < 0.05).

Accordingly, the results showed that the Alpha 4 variety (*S. bicolor*) possesses a significant Cu mg/kg^−1^ content within 10 mg/kg^−1^ in the endosperm (grits). This discovery contradicts the outcomes found among middle- and late-ripening-stage varieties.

#### 3.2.8. Iron

The HFTU process positively impacted Fe mg/kg^−1^ contents among all investigated sorghum inbred line races; however, the effect of the HFTU process fluctuated among endosperm (grits) and bran, specifically among the *S. bicolor* races, as is represented in Table 1 and Table 2, with a significance value of *p* < 0.05 in case of whole grains (ground). The findings showed that the male sterility line and the different ripping stages influenced the iron nutrient contents and accumulation. For example, in the case of *S. bicolor*, the Alpha 12 variety had the highest Fe mg/kg^−1^ content, around 147 mg/kg^−1^ on average. In contrast, Alpha 2, Alpha 4, and Alpha 11 varieties showed a reduction in Fe mg/kg^−1^ content compared with the endosperm to bran, within a percentage of 4%, 7%, and 2%, respectively (*p* > 0.05).

#### 3.2.9. Zinc

The findings indicated that Zn mg/kg^−1^ contents were enhanced after the HFTU process applications, as shown in Table 1 and Table 2. Furthermore, that was evidenced by higher zinc mineral accumulation in the husked grain (bran) than in whole grains (ground) within an average of 23.3 mg/kg^−1^ and 58.7 mg/kg^−1^, respectively. The findings are demonstrated in Table 1 and Table 2.

The positive effect of the 30 (S) and 80 (S) time units demonstrated variation ranges within an average of 53.1 mg/kg^−1^ and 64.4 mg/kg^−1^, respectively, and a ratio of 9.64%.

Moreover, Figure 1 and Figure 2 showed that 80 (S) time units significantly impacted the reconcentration of iron and zinc minerals in the diverse sorghum races (*S. bicolor* and Kafirin) and demonstrated a strong correlation between Fe mg/kg^−1^ and Zn mg/kg^−1^ with a coefficient of r = 0.517 and r = 0.893, respectively. These findings were supported by statistical significance (*p* < 0.05) in the case of *S. bicolor* and (*p* < 0.05) in the case of Kafirin bran, as shown in Table 2.

#### 3.2.10. Manganese

The influence of the HFTU process on Mn mg/kg^−1^ was observed by a good accumulation of Mn mg/kg^−1^ in the husked products within an average of 34.6 mg/kg^−1^ in the bran, compared to the whole grain (ground) within 17.7 mg/kg^−1^. The reported differences were attributed to sorghum race varieties. Conversely, the variation ratio derived from 80 (S) and 30 (S) time units was observed to be 10.13%, with a significance value of *p* < 0.05.

### 3.3. Evaluation of Color Profiles

The findings showed that the color of the studied varieties was affected by the HFTU process, with a duration of 30 (S) and 80 (S) time units. The impact of the HFTU process on the color features of the varieties showed a significant variance of *p* ≤ 0.05.

#### Color Characteristics Attributed to 30 (S) and 80 (S) Time Units

According to the revealed results of HFTU process on the color properties, the color change evaluations showed that the color profile characteristics were significantly influenced by 30 (S) and 80 (S) time units, Figure 3, supported with a significance value of *p* < 0.05. Specifically, the positive spectrum of color parameters in the endosperm (grits) was seen commencing at 30 (S) time unit.

The color changes of the bran, which occurred throughout 80 (S) time units, did not impact the varieties with brown pericarp color (Table 1 and Table 2) throughout the middle and late maturation phases, which was evidenced by the negative spectrum of a* (redness) at (−2). Furthermore, the decrease in darkness (L*) and brightness (Y) was noticed as a consequence of the Husking Fraction Time Unit HFTU process lasting for 80 time units (S). The spectrum of darkness ranged from 40 to 70, while the spectrum of brightness ranged from 35 to 50.

## 4. Discussion

The research showed that the HFTU process could fill the sorghum dehulling gap with a distinct separation milling method and contribute to nutritional content enhancements. The Husking Fraction Time Unit HFTU showed that the endosperm (grits) and bran had distinct nutritional profiles after the dehulling process, specifically when compared to whole grain (ground). These findings are consistent with previous studies [36,37]. The research emphasized the beneficial impact of the dry-fractionation process on enhancing the nutritional content of cereal grains ecologically and sustainably.

The revealed result of dry matter reconcentration (Table 1 and Table 2) was aligned with the study [38], which reported that the nutrient composition of sorghum grains can vary depending on the variety and year of cultivation, which can indirectly affect the dry matter content of the grains.

The sufficiency and accuracy of the HFTU technique results were evaluated according to the accumulation of protein level percentage in the bran and endosperm (grits); the results aligned to a previous study reported by [39], in which the authors explained the relationships between the protein content with the ripping stages, focusing on the protein content in the endosperm part among different maturity stage inbred lines. The accuracy and precision of the findings showed that Kafirin races have an abundant protein content (Table 2), ranging from 13 to 18%. These results aligned with previous reports [37,40], in which the authors reported that the highest total protein content was almost double in the Kafirin mutant line.

Additionally, the findings showed that the investigated sorghum races were distinguished by a high nutritional content. Specifically, the Kafirin race showed a strong correlation between the associated minerals, such as Ca:P, K:Na, and Fe:Zn attributed 80 (S) time units, with correlation coefficients of r = 0.884, r = 0.745, and r = 0.893, respectively, as were shown in Figure 2, making it advantageous for potential product development. These results aligned with [41], which emphasized the diverse impact of the employed processing techniques on the qualities of the nutritional contents. While the ripping phases impacted the iron (Fe) concentration, the study suggested a strong association with consequences for the grain’s hardness and the nutrients’ composition. According to the findings of the current study, the Alpha 4 variation of the *S. bicolor* races showed a rise in iron concentration. Combining the variety’s protein content with the observed increase in iron concentration resulted in associated nutrient contents. This study’s outcomes were supported by a previous report [42], which clarified the connection between iron content in inbred line varieties and genetic variability.

Notably, the observed color variations throughout grain maturation stages were related to the duration of 30 (S) and 80 (S) time units of the dehulling process [43], according to Figure 3, as explained by outcomes of the impact of the HFTU process on the color characterizations.

## 5. Conclusions

This research investigated a novel dehulling procedure for sorghum grain processing, specifically on sorghum inbred male sterility line of two sorghum race varieties. The aim was to establish a sustainable husking process and contribute to sorghum hybrid development programs by assessing the nutritional properties. The analysis demonstrated that racial diversity and pericarp color substantially influenced the nutritional composition of inbred line varieties; moreover, the findings referred to the influence of ripping times. The study found that setting the timers to 30 (S) and 80 (S) seconds significantly improved the husking process and led to distinct nutritional contents of each fraction milling of the studied varieties. Furthermore, the findings related to the Kafirin race demonstrate its potential as a promising raw material for edible plants that are rich in protein, highlighting its potential for use in food applications. Despite a significant variance based on the sorghum races, the findings did not show differences based on the type of the male sterility line: sterility line (A), maintainer line (B), and restorer line (R). On the other hand, the research suggests using processing techniques to augment the nutritional content to promote potential industrial sorghum products through the influence of the color profile of the HFTU process.

However, the findings can address the hardness test as a limitation, highlighting the necessity for a deeper understanding of how hardness affects the nutritional content of the husked endosperm (grits) and bran. In particular, this study reported the results related to the ripping stages. The researchers recommend involving the hardness test in further investigations of the seeds to comprehensively understand how nutrients build up from the bran and endosperm after the husking and fractions separation processes. In addition, further evaluation of the revealed protein content properties and conducting rheological property testing can enhance and promote the HFTU technique to optimize the nutritional profile of the examined sorghum race varieties.

## Figures and Tables

**Figure 1 foods-13-01100-f001:**
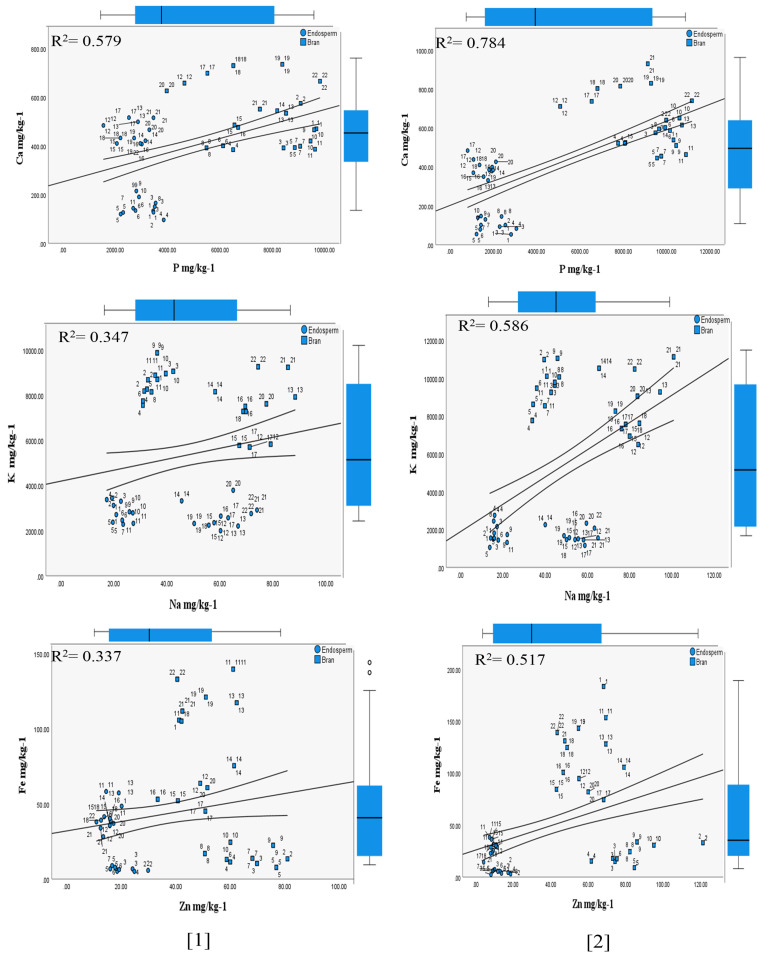
Relationships between the most interacted mineral contents of the *S. bicolor* races inferred from the HFTU process, *p* ≤ 0.05.The Numbers: [1] correlation based on 30 (S) time units, [2] correlation based on 80 (S) time units; the linear correlation analysis was performed based on mean values, with a 95% confidence level.

**Figure 2 foods-13-01100-f002:**
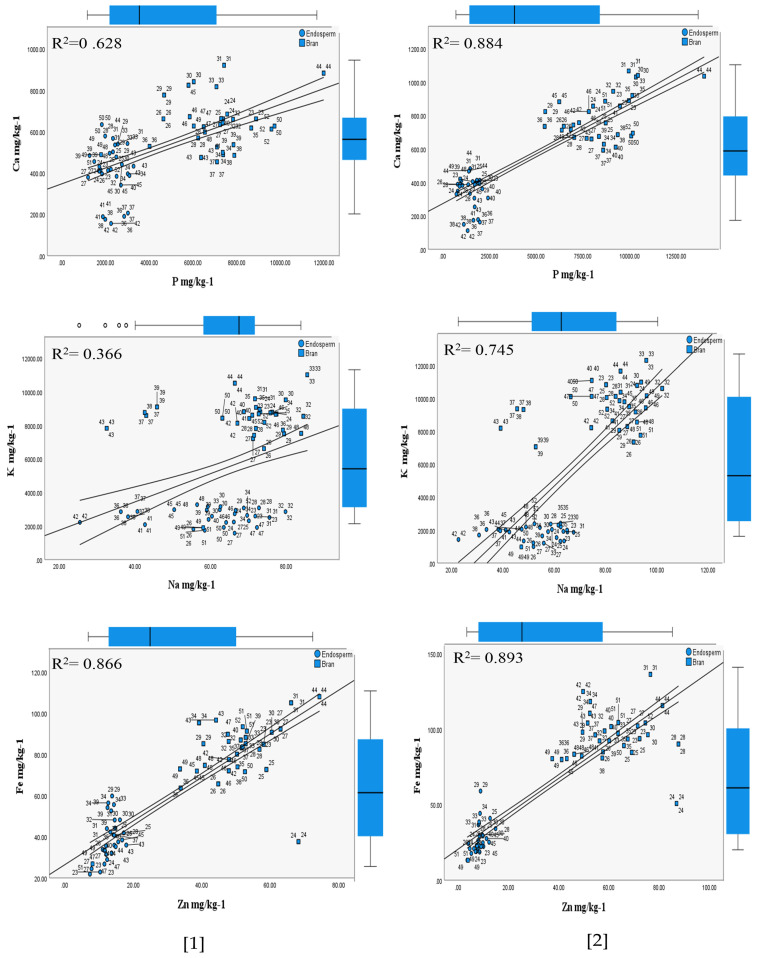
Relationships between the most interacting mineral contents of the Kafirin races inferred from the HFTU process, *p* ≤ 0.05. The numbers: [1] correlation based on 30 (S) time units, [2] correlation based on 80 (S) time units; the linear correlation analysis was performed based on mean values, with a 95% confidence level.

**Figure 3 foods-13-01100-f003:**
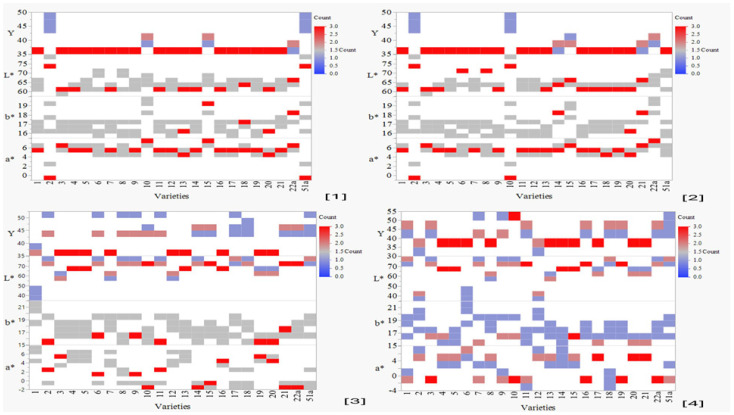
Features of the color profile of the investigated inbred line varieties based on the endosperm (grits) and bran, *p* ≤ 0.05. Numbers references are as follows: [1] endosperm color properties in case of 30 (S), [2] bran endosperm color properties in case of 30 (S), [3] endosperm color properties in case of 80 (S), and [4] bran color properties in case of 80 (S). 22a refers to Alpha 22 variety (Kafirin races) and 51a refers to Alpha 51 variety (Kafirin races).

**Table 1 foods-13-01100-t001:** Descriptive analysis of the nutritional contents of diverse *S. bicolor* race varieties on male sterility lines.

Codes	Processed Grains	Pericarp Color	D.M g/kg^−1^	Protein g/kg^−1^	P mg/kg^−1^	K mg/kg^−1^	S mg/kg^−1^	Ca mg/kg^−1^	Mg mg/kg^−1^	Cu mg/kg^−1^	Fe mg/kg^−1^	Zn mg/kg^−1^	Na mg/kg^−1^	Mn mg/kg^−1^
1	1		986 ± 0.03	22 ± 0.13	4201 ± 36.1	3932 ± 2926.3	1727 ± 12.4	174 ± 6.1	2643 ± 0.2	4 ± 0.2	83 ± 0.1	29 ± 0.03	23 ± 0.3	16 ± 1.1
	2	Brown	987 ± 0.47	8 ± 0.64	2630 ± 201.7	2965 ± 166.9	1372 ± 54.3	114 ± 14.3	1721 ± 447.5	5 ± 0.7	25 ± 0.7	24 ± 6.8	18 ± 2.1	11 ± 2.8
	3		987 ± 0.47	14 ± 0.21	9993 ± 347.8	9404 ± 771.8	1889 ± 130.2	505 ± 37.7	4722 ± 166.3	8 ± 2.8	145 ± 42.6	55 ± 14.7	34 ± 2.4	37 ± 13.6
2	1		890 ± 0.13	23 ± 0.03	3877 ± 81.2	3650 ± 0.41	1517 ± 202.1	176 ± 8.0	2313 ± 0.1	6 ± 0.30	39 ± 0.3	31 ± 0.1	21 ± 0.3	15 ± 2.2
	2	White	990 ± 0.05	16 ± 0.26	3005 ± 496.7	2929 ± 87.4	1591 ± 32.6	93 ± 43.7	1645 ± 541.6	3 ± 0.70	39 ± 10.5	15 ± 5.1	18 ± 3.6	8 ± 3.0
	3		990 ± 0.05	13 ± 0.10	9525 ± 492.9	9830 ± 1260.4	1705 ± 95.3	607 ± 36.3	4951 ± 458.3	13 ± 4.4	23 ± 10.6	101 ± 21.9	36 ± 3.6	57 ± 20.7
3	1		991 ± 0.04	12 ± 0.60	3712 ± 14.2	2983 ± 0.4	1760 ± 67.2	261.3 ± 12.1	2344 ± 0.3	4 ± 0.2	35 ± 0.1	28 ± 0.1	21 ± 1.0	15 ± 2.2
	2	Brown	991 ± 0.04	9 ± 0.58	2396 ± 119.2	3729 ± 469.0	1579 ± 142.6	122 ± 31.2	1645 ± 468.8	3 ± 0.8	26 ± 0.6	18 ± 6.7	20 ± 2.9	9 ± 3.1
	3		991 ± 0.04	18 ± 0.55	9525 ± 492.9	9830 ± 1260.4	1705 ± 95.3	607 ± 36.3	4951 ± 458.3	13 ± 4.4	23 ± 10.6	101 ± 21.9	36 ± 3.9	57 ± 20.7
4	1		990 ± 0.05	20 ± 0.11	4185 ± 128.0	4047 ± 11.0	1555 ± 20.3	243 ± 11.2	2350 ± 0.3	3 ± 0.5	30 ± 0.8	30 ± 0.1	27 ± 0.5	14 ± 0.2
	2	Brown	990 ± 0.05	15 ± 0.02	8452 ± 1527.8	9565 ± 208.01	1412 ± 67.5	90 ± 7.3	2906 ± 507.6	10 ± 0.2	94 ± 1.1	22 ± 11.3	17 ± 0.6	33 ± 2.4
	3		990 ± 0.05	12 ± 0.09	8953 ± 579.6	9345 ± 319.1	1703 ± 214.2	484 ± 100.8	4312 ± 233.3	9 ± 1.5	13 ± 2.5	72 ± 2.4	44 ± 1.2	36 ± 0.1
5	1		98.4 ± 0.28	17 ± 0.41	2155 ± 125.1	2289 ± 192.2	1437 ± 234.0	192 ± 118.5	1399 ± 248.8	3 ± 0.2	39 ± 15.2	24 ± 7.0	23 ± 0.3	16 ± 1.1
	2	Brown	991 ± 0.06	8 ± 0.02	1712 ± 539.8	2213 ± 170.5	1293 ± 49.4	87 ± 35.7	1525 ± 97.7	3 ± 0.5	30 ± 3.0	12 ± 4.2	17 ± 3.1	8 ± 0.9
	3		991 ± 0.05	12 ± 0.09	2128 ± 407.8	3160 ± 663.0	1716 ± 85.5	453 ± 75.3	2342 ± 277.2	5 ± 1.7	26 ± 0.1	61 ± 0.9	32 ± 1.6	19 ± 1.0
6	1		980 ± 0.04	20 ± 0.02	3505 ± 97.1	5326 ± 0.2	1612 ± 164.0	264 ± 21.8	2384 ± 0.2	4 ± 0.1	46 ± 0.1	26 ± 0.2	22 ± 9.7	25 ± 6.5
	2	White	991 ± 0.07	6 ± 0.02	2081 ± 756.2	2450 ± 9.3	1482 ± 99.3	107 ± 30.3	143 ± 521.4	3 ± 0.5	24 ± 0.8	16 ± 2.5	21 ± 2.8	9 ± 3.2
	3		991 ± 0.06	12 ± 0.10	9199 ± 383.6	8442 ± 198.0	1688 ± 16.9	419 ± 29.1	4142 ± 94.2	10 ± 1.1	28 ± 0.6	81 ± 4.2	33 ± 1.0	32 ± 1.4
7	1	Brown	894 ± 0.02	13 ± 0.01	2521 ± 357.1	2514 ± 180.8	1522 ± 246.2	323 ± 153.1	1298 ± 336.2	5 ± 1.1	26 ± 2.2	25 ± 4.6	27 ± 0.1	16 ± 0.1
	2		991 ± 0.05	12 ± 0.02	1861 ± 481.1	1397 ± 129.4	1251 ± 23.2	114 ± 13.6	1351 ± 342.2	2 ± 0.2	18 ± 0.8	13 ± 3.5	20 ± 4.3	11 ± 3.4
	3		991 ± 0.05	19 ± 0.03	8130 ± 2209.8	8606 ± 947.0	1815 ± 237.4	494 ± 29.1	4542 ± 617.7	10 ± 0.5	15 ± 2.4	67 ± 9.0	34 ± 2.7	40 ± 6.3
8	1		984 ± 0.07	20 ± 0.02	3733 ± 112.3	4041 ± 0.24	1312 ± 75.0	296 ± 1.5	2516 ± 0.8	6 ± 0.4	28 ± 0.3	25 ± 0.23	29 ± 12.5	21 ± 7.4
	2	White	991 ± 0.06	9 ± 0.32	2958 ± 638.7	2945 ± 542.9	1141 ± 62.8	155 ± 10.9	1735 ± 338.5	3 ± 0.5	26 ± 0.6	15 ± 2.8	18 ± 1.9	15 ± 3.5
	3		991 ± 0.06	11 ± 0.11	9199 ± 383.6	8442 ± 198.0	1688 ± 16.9	419 ± 29.1	4142 ± 94.2	10 ± 1.1	28 ± 0.6	81 ± 4.2	33 ± 1.0	32 ± 1.4
9	1		985 ± 0.08	12 ± 0.001	2742 ± 93.4	2547 ± 447.0	1347 ± 254.0	274 ± 3.8	1714 ± 222.4	2 ± 0.39	31 ± 12.9	19 ± 7.9	27 ± 0.1	24 ± 1.1
	2	White	991 ± 0.06	19 ± 0.33	2215 ± 644.5	2782 ± 55.9	1192 ± 48.1	173 ± 46.3	1316 ± 429.0	3 ± 0.3	26 ± 0.4	16 ± 4.0	24 ± 2.0	10 ± 2.6
	3		990 ± 0.02	10 ± 0.01	7564 ± 2281.3	4105 ± 1054.7	1689 ± 213.7	427 ± 31.1	4065 ± 590.8	13 ± 2.3	20 ± 4.0	66 ± 17.2	40 ± 6.8	55 ± 11.4
10	1	White	983 ± 0.23	18 ± 0.003	2644 ± 179.1	4617 ± 311.0	1586 ± 415.4	319 ± 298.5	1578 ± 254.7	4 ± 1.0	35 ± 12.0	23 ± 7.8	28 ± 57.1	19 ± 7.9
	2		990 ± 0.02	14 ± 0.35	2165 ± 644.5	1659 ± 138.1	1402 ± 166.2	169 ± 23.6	782 ± 2.2	1 ± 0.4	17 ± 1.1	13 ± 4.6	22 ± 6.1	9 ± 3.6
	3		990 ± 0.02	15 ± 0.39	10013 ± 464.3	10455 ± 646.6	1652 ± 47.4	494 ± 111.6	4744 ± 363.6	14 ± 2.2	28 ± 5.5	81 ± 5.5	41 ± 5.2	53 ± 6.6
11	1		983 ± 0.23	8 ± 0.001	2390 ± 372.1	5261 ± 197.0	1562 ± 280	335 ± 173.0	1712 ± 155.6	4 ± 0.5	36 ± 8.4	23 ± 3.6	30 ± 14.7	13 ± 2.9
	2	Brown	990 ± 0.02	15.5 ± 0.38	1981 ± 771.7	2820 ± 544.0	1331 ± 103.0	142 ± 2.4	1176 ± 496.2	2 ± 0.5	48 ± 11.0	11 ± 3.8	25 ± 2.9	8 ± 1.9
	3		990 ± 0.02	10 ± 0.26	10,019 ± 622.0	10,868 ± 94.9	1931 ± 95.9	488 ± 24.2	4045 ± 191.8	11 ± 0.2	27 ± 3.3	78 ± 19.1	42 ± 2.7	61 ± 4.6
12	1		986 ± 0.01	19 ± 0.50	2790 ± 1824.1	5524 ± 456.8	1044 ± 237.0	534 ± 119.5	1942 ± 942.3	5 ± 2.5	37 ± 6.9	22 ± 7.0	34 ± 10.8	20 ± 4.4
	2	Brown	991 ± 0.06	12 ± 0.34	1315 ± 255.3	2753 ± 266.4	901 ± 35.5	461 ± 24.8	929 ± 163.5	4 ± 0.2	33 ± 2.4	13 ± 2.6	58 ± 2.5	8 ± 1.9
	3		991 ± 0.06	18 ± 0.08	10259 ± 690.5	9067 ± 205.5	1688 ± 82.1	536 ± 126.9	4620 ± 168.1	8 ± 0.5	147 ± 7.5	65 ± 4.5	39 ± 3.9	37 ± 4.4
13	1	Brown	985 ± 0.04	12 ± 0.01	3589 ± 114	2936 ± 0.38	1244 ± 0.2	509 ± 2.2	2226 ± 0.2	6 ± 0.3	44 ± 0.4	24 ± 0.2	66 ± 12.8	15 ± 0.7
	2		990 ± 0.05	9 ± 0.53	2307 ± 611.3	1328 ± 136.6	1086 ± 47.6	415 ± 91.8	1424 ± 401.3	3 ± 0.1	44 ± 14.1	14 ± 5.2	62 ± 4.7	9 ± 2.4
	3		990 ± 0.02	14 ± 0.59	4860 ± 240.1	6163 ± 368.7	1335 ± 44.8	425 ± 42.4	2994 ± 99.6	10 ± 1.9	79 ± 17.0	52 ± 3.3	82 ± 2.6	29 ± 5.5
14	1		985 ± 0.13	8 ± 0.001	3740 ± 520.9	4252 ± 2.7	1150 ± 77.2	274 ± 4.6	1536 ± 0.2	4 ± 0.4	41 ± 0.20	17 ± 0.4	73 ± 1.1	14 ± 0.1
	2	White	991 ± 0.02	0.93 ± 0.16	2546 ± 662.8	3787 ± 514.5	1057 ± 66.3	411 ± 11.7	1068 ± 290.3	2 ± 0.1	31 ± 5.5	13 ± 3.4	43 ± 2.9	9 ± 2.9
	3		990 ± 0.02	14 ± 0.12	9607 ± 1194.0	11949 ± 30.2	1622 ± 117.4	684 ± 29.3	4398 ± 520.0	14 ± 0.9	122 ± 5.9	66 ± 3.8	91 ± 3.2	40 ± 0.2
15	1		981 ± 0.11	15 ± 0.02	3667 ± 86.1	3443 ± 126.8	1095 ± 18.6	419 ± 20.0	1883 ± 1.0	3 ± 0.4	50 ± 0.1	22 ± 0.1	60 ± 8.3	15 ± 1.8
	2	Brown	990 ± 0.01	8 ± 0.42	2279 ± 804.6	3056 ± 451.6	885 ± 56.9	379 ± 31.5	1139 ± 395.0	2 ± 0.2	31 ± 8.3	12 ± 4.0	57 ± 3.3	9 ± 3.0
	3		990 ± 0.02	19 ± 0.12	7319 ± 841.9	6363 ± 636.1	1344 ± 149.5	574 ± 32.0	3079 ± 261.6	8 ± 1.7	68 ± 17.7	42 ± 1.2	74 ± 6.9	24 ± 3.4
16	1		985 ± 0.36	7.8 ± 0.003	3529 ± 1038.1	3308 ± 637.2	975 ± 94.3	611 ± 94.5	1747 ± 547.3	3 ± 0.9	34 ± 5.3	18 ± 4.1	63 ± 0.8	18 ± 1.2
	2	Brown	990 ± 0.06	13 ± 0.15	1664 ± 937.6	2365 ± 221.9	1154 ± 3.8	500 ± 17.2	788 ± 461.6	3 ± 0.7	21 ± 7.5	9 ± 4.9	61 ± 2.3	5 ± 1.3
	3		991 ± 0.02	15 ± 0.09	7378 ± 752.8	7421 ± 89.3	1325 ± 118.2	506 ± 21.7	3404 ± 81.7	4 ± 0.9	77 ± 26.1	40 ± 7.3	73 ± 3.6	26 ± 5.7
17	1		983 ± 0.15	13 ± 0.1	4285 ± 383.3	3308 ± 637.2	1154 ± 293.6	544 ± 174.2	1747 ± 547.3	4 ± 0.6	49 ± 2.7	25 ± 2.8	59 ± 9.8	16 ± 2.0
	2	Brown	991 ± 0.01	15 ± 0.39	1777 ± 459.9	2354 ± 123.3	1118 ± 136.8	423 ± 11.7	980 ± 249.5	2 ± 0.5	34 ± 5.7	10 ± 2.4	53 ± 3.0	12 ± 2.0
	3		990 ± 0.07	15 ± 0.25	6023 ± 558.9	6628 ± 1017.2	1131 ± 81.5	499 ± 24.9	2713 ± 483.6	10 ± 1.8	60 ± 16.0	60 ± 9.4	75 ± 3.7	18 ± 1.4
18	1		986 ± 0.01	12 ± 0.11	3723 ± 249.1	3526 ± 500.0	1236 ± 64.5	478 ± 3.1	1828 ± 42.6	4 ± 0.2	51 ± 0.6	22 ± 2.3	61 ± 6.9	15 ± 2.4
	2	White	990 ± 0.01	12 ± 0.10	2323 ± 433.9	1995 ± 349.4	935 ± 53.6	411 ± 37.6	1149 ± 196.8	3 ± 0.2	35 ± 6.2	14 ± 2.3	50 ± 0.7	10 ± 1.8
	3		990 ± 0.05	14 ± 0.41	6652 ± 161.1	7449 ± 185.4	1430 ± 34.8	718 ± 21.2	3249 ± 215.9	6 ± 0.3	115 ± 10.8	45 ± 3.6	77 ± 18.6	36 ± 11.8
19	1		990 ± 0.60	10 ± 0.10	3001 ± 752.4	3521 ± 1008.2	1308 ± 177.1	265 ± 29.4	1684 ± 334.9	3 ± 0.6	40 ± 0.1	17 ± 5.0	66 ± 5.0	15 ± 3.4
	2	Brown	990 ± 0.07	15 ± 0.30	2629 ± 907.6	2234 ± 744.3	757 ± 84.5	433 ± 64.4	1198 ± 362.6	2 ± 0.4	28 ± 6.0	10 ± 2.4	69 ± 4.9	7 ± 1.3
	3		99.12 ± 0.01	17 ± 0.40	8826 ± 492.5	7765 ± 542.1	1331 ± 85.8	767 ± 40.5	3434 ± 146.2	8 ± 0.7	132 ± 12.2	53 ± 2.0	71 ± 1.8	35 ± 4.9
20	1		984 ± 0.01	12 ± 0.17	3546 ± 92.1	3760 ± 1416.0	1022 ± 49.6	534 ± 22.2	2182 ± 1103.3	3 ± 2.2	34 ± 0.2	16 ± 6.1	79 ± 7.1	17 ± 10.1
	2	Brown	990 ± 0.05	15 ± 0.47	2706 ± 650.5	3061 ± 795.8	757 ± 84.5	433 ± 64.4	1198 ± 362.6	2 ± 0.4	28 ± 6.0	10 ± 2.4	69 ± 4.9	7 ± 1.3
	3		990 ± 0.02	12 ± 0.53	5909 ± 133.4	8330 ± 786.5	1308 ± 32.4	783 ± 52.6	3406 ± 293.9	10 ± 2.3	71 ± 11.5	56 ± 4.5	81 ± 3.3	30 ± 5.6
21	1		984 ± 0.06	10 ± 0.01	3792 ± 123.2	4014 ± 11.3	1179 ± 19.0	426 ± 4.0	1523 ± 0.0	4 ± 0.1	48 ± 0.2	15 ± 0.1	7 ± 0.2	16 ± 0.1
	2	White	990 ± 0.05	15 ± 0.39	2471 ± 530.7	2412 ± 373.3	1119 ± 136.8	398 ± 14.5	1044 ± 211.1	3 ± 0.1	35 ± 3.6	10 ± 0.6	68 ± 4.6	5 ± 1.7
	3		990 ± 0.02	12 ± 0.53	83,322 ± 878.9	10,177 ± 1035.2	1206 ± 160.0	721 ± 130.8	3856 ± 504.6	6 ± 0.5	121 ± 10.6	45 ± 2.8	93 ± 8.2	40 ± 2.1
Ground grains	-	-	0.23	<0.001	<0.001	<0.001	<0.001	<0.001	0.28	<0.001	<0.001	0.001	0.02	<0.001
Endosperm	-	-	<0.001	<0.001	0.79	0.81	<0.001	<0.001	0.05	<0.001	0.43	0.88	0.28	0.54
Bran	-	-	<0.001	0.52	0.36	<0.001	0.03	0.55	<0.001	0.54	0.53	0.32	0.04	0.05

Numbers 1–11 and 13 were on the A (sterile line), code 12 on the B (maintaining line), and codes 14–21 were on R (restoring line); numbers refer to 1: whole grains (ground), 2: endosperm (grits), 3: bran. Values presented as mean ± SD, values of the mineral mg/kg^−1^ were analyzed based on three replication readings, while the dry matter and protein % were analyzed based on two replication readings, *p* ≤ 0.05.

**Table 2 foods-13-01100-t002:** Descriptive analysis of the nutritional contents of diverse Kafirin race varieties on male sterility line.

Codes	Processed Grains	Pericarp Color	D.M g/kg^−1^	Protein g/kg^−1^	P mg/kg^−1^	K mg/kg^−1^	S mg/kg^−1^	Ca mg/kg^−1^	Mg mg/kg^−1^	Cu mg/kg	Fe mg/kg^−1^	Zn mg/kg^−1^	Na mg/kg^−1^	Mn mg/kg^−1^
22	1	Brown	981 ± 0.06	13 ± 0.01	2263 ± 306.2	4642 ± 375.2	1241 ± 370.0	661 ± 248.6	1159 ± 890.0	7 ± 0.1	26 ± 3.5	21 ± 9.7	74 ± 7.9	19 ± 6.0
	2		991 ± 0.05	9 ± 0.07	1654 ± 280.7	2239 ± 386.0	757 ± 84.5	409 ± 4.5	885 ± 154.2	2 ± 0.4	23 ± 0.3	10 ± 0.1	68 ± 4.6	10 ± 0.9
	3		990 ± 0.02	13 ± 0.53	10486 ± 737.6	9869 ± 673.8	1462 ± 144.0	741 ± 208.1	3585 ± 245.2	8 ± 0.1	136 ± 3.4	42 ± 1.5	78 ± 4.3	43 ± 3.0
23	1	White	986 ± 0.03	11 ± 0.10	2762 ± 184.2	3012 ± 797.5	815 ± 192.0	475 ± 132.0	1450 ± 416.4	4 ± 1.0	28 ± 5.3	24 ± 7.9	64 ± 8.9	23 ± 6.2
	2		991 ± 0.49	14 ± 0.22	1343 ± 467.4	1891 ± 381.2	715 ± 53.9	412 ± 34.4	711 ± 248.8	3 ± 0.5	41 ± 0.4	9 ± 3.3	64 ± 2.8	6 ± 1.1
	3		993 ± 0.32	10 ± 0.39	7802 ± 266.5	9784 ± 1094.4	1728 ± 166.5	775 ± 124.3	3718 ± 549.5	16 ± 1.6	44 ± 7.2	78 ± 10.2	84 ± 8.7	21 ± 3.9
24	1	White	987 ± 0.98	19 ± 0.12	3741 ± 1127.1	4660 ± 2206.3	1000 ± 32.2	523 ± 133.3	1395 ± 176.4	5 ± 0.1	30 ± 9.9	22 ± 4.1	70 ± 5.6	17 ± 9.9
	2		990 ± 3.45	9.3 ± 0.12	2232 ± 280.1	2113 ± 215.9	989 ± 34.5	431 ± 36.1	1318 ± 146.4	3 ± 0.2	27 ± 7.0	8 ± 3.5	67 ± 3.5	13 ± 2.6
	3		991 ± 0.44	8 ± 0.02	8046 ± 737.1	9362 ± 744.1	1226 ± 135.3	770 ± 93.8	3491 ± 125.0	11 ± 1.0	79 ± 6.6	64 ± 5.5	77 ± 4.2	37 ± 3.6
25	1	White	984 ± 0.24	14 ± 0.02	3105 ± 66.0	3234 ± 170.8	1046 ± 16.0	554 ± 0.5	1611 ± 0.7	3 ± 0.9	51 ± 0.1	23 ± 0.1	64 ± 0.0	12 ± 0.1
	2		991 ± 0.01	14 ± 0.01	1317 ± 570.9	1406 ± 449.9	804 ± 54.1	389 ± 6.1	703 ± 315.5	3 ± 0.5	24 ± 0.1	6 ± 2.0	54 ± 1.9	8 ± 0.9
	3		991 ± 0.07	16 ± 1.3	5561 ± 1002.1	6977 ± 401.5	1091 ± 111.8	708 ± 51.6	2842 ± 389.7	10 ± 0.7	75 ± 10.6	51 ± 7.0	85 ± 9.2	26 ± 3.9
26	1	White	985 ± 0.08	13 ± 0.04	2453 ± 313.1	3253 ± 1483.5	996 ± 202.0	4438 ± 123.3	2718 ± 949.4	4 ± 0.1	54 ± 0.4	27 ± 2.1	66 ± 8.4	22 ± 8.2
	2		990 ± 0.12	11 ± 0.05	999 ± 218.6	1390 ± 201.2	877 ± 75.2	370 ± 90.1	517 ± 122.6	3 ± 0.8	35 ± 6.0	6 ± 2.0	61 ± 5.7	13 ± 0.2
	3		993 ± 0.02	20 ± 0.77	7467 ± 249.9	7749 ± 584.7	1182 ± 190.3	698 ± 40.3	3253 ± 250.8	11 ± 1.2	97 ± 4.2	67 ± 4.6	80 ± 9.4	35 ± 1.4
27	1	White	985 ± 0.08	132 ± 0.10	2069 ± 11.0	3253 ± 1483.5	1160 ± 13.0	609 ± 0.9	1264 ± 0.1	4 ± 1.3	29 ± 0.0	16 ± 0.1	79 ± 1.0	17 ± 0.1
	2		990 ± 0.03	118 ± 0.18	1670 ± 338.6	2690 ± 429.4	1048 ± 56.9	476 ± 98.7	1007 ± 198.5	4 ± 0.5	59 ± 0.3	12 ± 2.5	67 ± 6.1	9 ± 0.34
	3		991 ± 0.05	10.0 ± 0.50	6635 ± 441.6	8964 ± 1260.2	1289 ± 103.2	648 ± 15.7	3383 ± 360.7	10 ± 1.5	92 ± 16.8	72 ± 16.8	82 ± 3.2	35 ± 4.0
28	1	White	983 ± 0.01	13 ± 0.10	2546 ± 392.2	2589 ± 291.0	1106 ± 349.1	599 ± 233.7	1434 ± 389.2	2 ± 0.3	40 ± 10.2	35 ± 24.2	70 ± 9.5	28 ± 0.8
	2		990 ± 0.30	12 ± 0.31	2230 ± 358.5	2459 ± 292.4	896 ± 21.0	450 ± 71.1	1070 ± 210.0	3 ± 0.4	41 ± 7.7	11 ± 2.7	64 ± 2.0	7 ± 0.9
	3		991 ± 0.08	21 ± 0.78	5079 ± 440.9	7773 ± 306.8	1724 ± 70.4	619 ± 54.7	2736 ± 106.0	10 ± 0.3	93 ± 17.2	45 ± 4.9	86 ± 7.9	42 ± 3.8
29	1	White	981 ± 0.01	11 ± 0.10	2766 ± 219.3	3043 ± 681.0	1262 ± 327	592 ± 220.1	1218 ± 217.3	4 ± 0.8	40 ± 7.6	37 ± 29.7	65 ± 7.1	24 ± 4.7
	2		990 ± 0.04	14 ± 0.14	2356 ± 425.2	2668 ± 330.5	1066 ± 42.6	432 ± 55.9	1306 ± 142.3	3 ± 0.5	40 ± 3.9	16 ± 0.6	61 ± 2.0	7 ± 1.7
	3		990 ± 0.02	17 ± 1.12	8184 ± 2356.8	10,249 ± 799.8	1594 ± 121.3	7100 ± 26.2	3411 ± 354.9	12 ± 1.3	121 ± 17.2	68 ± 8.2	79 ± 7.7	30 ± 2.4
30	1	White	981 ± 1.01	7.0 ± 0.05	4299 ± 317.9	2704 ± 175.9	892 ± 212.1	566 ± 94.2	1464 ± 311.8	3 ± 0.7	56 ± 27.5	16 ± 12.0	77 ± 8.9	23 ± 7.4
	2		990 ± 0.20	14 ± 0.15	1946 ± 528.2	2188 ± 352.5	765 ± 48.5	483 ± 55.5	1045 ± 250.1	3 ± 0.5	40 ± 36.5	10 ± 2.3	64 ± 8.1	8 ± 2.2
	3		991 ± 0.05	15 ± 0.003	8682 ± 1384.8	9980 ± 435.5	1745 ± 148.0	935 ± 103.4	3995 ± 49.0	14 ± 2.0	92 ± 6.8	71 ± 5.6	93 ± 9.5	30 ± 1.9
31	1	White	980 ± 0.46	10 ± 0.01	2060 ± 540.1	6371 ± 452.4	1140 ± 316.2	518 ± 217.0	1674 ± 335.0	4 ± 0.8	59 ± 27.4	22 ± 6.9	78 ± 10.9	17 ± 8.1
	2		991 ± 0.03	13 ± 0.30	1980 ± 574.6	2430 ± 467.7	954 ± 78.5	385 ± 80.9	1038 ± 268.0	3 ± 0.62	43 ± 5.7	11 ± 3.4	64 ± 8.1	8 ± 2.2
	3		991 ± 0.07	19 ± 0.36	9473 ± 711.0	9589 ± 1128.2	1281 ± 184.2	993 ± 80.1	3412 ± 204.3	10 ± 0.8	93 ± 5.0	53 ± 5.6	91 ± 5.6	28 ± 4.1
32	1	White	987 ± 0.16	16 ± 0.03	2780 ± 864.0	3709 ± 1972.9	1016 ± 153.0	485 ± 6.8	1483 ± 395.9	4 ± 1.5	45 ± 6.1	28 ± 12.6	67 ± 9.1	23 ± 7.0
	2		991 ± 0.03	12 ± 0.08	2402 ± 655.9	2596 ± 614.7	933 ± 89.0	450 ± 71.1	1165 ± 309.9	3 ± 0.5	50 ± 6.6	11 ± 3.3	61 ± 1.9	7 ± 1.4
	3		991 ± 0.01	13 ± 0.23	8755 ± 1848.8	11658 ± 696.1	1597 ± 137.5	802 ± 156.3	4074 ± 233.3	12 ± 1.7	107 ± 12.6	58 ± 5.8	81 ± 5.2	38 ± 4.6
33	1	White	980 ± 0.21	12 ± 1.06	2995 ± 83.8	5270 ± 349.1	1188 ± 292.0	463 ± 143.3	1477 ± 333.5	4 ± 1.5	46 ± 4.7	26 ± 2.9	67 ± 8.0	17 ± 1.8
	2		990 ± 0.11	12 ± 0.19	2566 ± 491.3	2269 ± 399.1	1013 ± 507.0	382 ± 24.5	1311 ± 215.5	3 ± 0.2	33 ± 9.5	11 ± 2.2	64 ± 6.6	9 ± 1.4
	3		980 ± 0.10	16 ± 0.35	7998 ± 682.2	9301 ± 604.5	1263 ± 127.6	928 ± 122.3	3132 ± 90.6	7 ± 1.1	87 ± 7.3	46 ± 4.3	81 ± 9.4	26 ± 1.9
34	1	White	982 ± 0.19	10 ± 0.01	3020 ± 943.2	2894 ± 840.2	998 ± 159.0	450 ± 76.4	1905 ± 646.2	4 ± 1.6	41 ± 14.9	26 ± 3.4	70 ± 9.0	15 ± 2.1
	2		990 ± 0.05	19 ± 0.03	1974 ± 291.2	2677 ± 287.9	900 ± 45.8	365 ± 87.0	1084 ± 154.7	2 ± 0.4	50 ± 6.6	12 ± 1.9	65 ± 2.2	8 ± 2.0
	3		990 ± 0.03	18 ± 0.15	9399 ± 810.6	9283 ± 226.2	1499 ± 4.1	558 ± 76.5	3837 ± 126.3	13 ± 1.0	72 ± 8.9	59 ± 9.4	85 ± 6.8	27 ± 2.7
35	1	White	987 ± 0.02	16 ± 0.30	3337 ± 523.2	2894 ± 840.2	1051 ± 237.1	278 ± 159.4	1544 ± 326.3	5 ± 2.1	38 ± 13.6	22 ± 7.6	38 ± 5.0	14 ± 4.0
	2		989 ± 0.01	12 ± 0.34	2368 ± 526.1	2447 ± 448.5	907 ± 50.4	186 ± 11.2	1144 ± 257.1	2 ± 0.4	39 ± 14.4	11 ± 0.3	35 ± 1.4	8 ± 2.0
	3		989 ± 0.50	17 ± 0.09	4729 ± 790.3	8445 ± 792.6	1279 ± 135.4	768 ± 165.6	2656 ± 187.7	8 ± 0.9	90 ± 7.1	37 ± 3.9	44 ± 1.3	26 ± 2.9
36	1	White	980 ± 0.01	14 ± 0.12	2692 ± 598.4	3295 ± 1472.3	1123 ± 236.6	288 ± 178	2135 ± 869.4	3 ± 0.5	36 ± 11.6	22 ± 7.6	41 ± 2.4	17 ± 1.5
	2		990 ± 0.06	8.5 ± 0.70	2501 ± 568.3	2457 ± 450.9	987 ± 63.1	179 ± 21.4	1153 ± 218.3	2 ± 0.2	32 ± 9.9	12 ± 2.6	39 ± 1.3	7 ± 1.6
	3		99.0 ± 0.23	16 ± 0.01	7839 ± 812.7	8976 ± 429.2	1385 ± 97.1	631 ± 113.1	3377 ± 124.9	12 ± 1.4	77 ± 3.9	53 ± 1.6	45 ± 2.9	28 ± 0.9
37	1	White	983 ± 0.01	11 ± 0.00	2375 ± 1462.0	4846 ± 381.7	1062 ± 180.1	291 ± 22.0	1800 ± 902.3	5 ± 4.2	31 ± 12.3	22.6.6	38 ± 7.8	19 ± 6.0
	2		991 ± 0.03	13 ± 0.07	1554 ± 471.1	2122 ± 470.0	958 ± 80.6	214 ± 33.0	730 ± 264.5	2 ± 1.0	32 ± 12.7	11 ± 3.8	34 ± 4.0	7 ± 2.1
	3		990 ± 0.54	12 ± 0.25	8624 ± 809.5	9042 ± 296.9	1437 ± 93.3	523 ± 75.7	3198 ± 190.0	10 ± 0.6	89 ± 3.9	54 ± 1.6	49 ± 3.8	27 ± 0.5
38	1	White	987 ± 0.01	73 ± 0.02	3254 ± 361.1	2681 ± 271.3	960 ± 165.2	518 ± 122.0	1705 ± 237.8	5 ± 1.3	28 ± 1.8	21 ± 7.4	62 ± 7.2	18 ± 8.7
	2		990 ± 0.06	12 ± 0.88	1130 ± 148.5	202 ± 411.0	856 ± 21.4	431 ± 40.0	721 ± 25.3	3 ± 0.5	36 ± 19.2	10 ± 2.8	57 ± 2.2	6 ± 4.0
	3		990 ± 0.76	18 ± 0.25	8105 ± 277.5	8076 ± 1118.7	1391 ± 176.4	585 ± 110.0	3107 ± 387.9	10 ± 0.8	94 ± 7.3	56 ± 3.9	72 ± 3.6	34 ± 1.0
39	1	White	982 ± 0.21	13 ± 0.05	4348 ± 274.0	4448 ± 3570.5	1113 ± 169.0	403 ± 9.9	1362 ± 173.9	4 ± 0.4	36 ± 8.6	20 ± 4.3	62 ± 8.3	13 ± 4.5
	2		991 ± 0.03	10 ± 0.15	2753 ± 367.9	2369 ± 233.3	1007 ± 23.3	312 ± 78.8	1329 ± 184.8	3 ± 0.6	32 ± 4.5	13 ± 1.7	57 ± 3.2	9 ± 1.0
	3		993 ± 0.42	17 ± 0.11	8304 ± 1036.8	9951 ± 1241.5	1242 ± 120.6	605 ± 76.0	3772 ± 189.2	12 ± 0.5	88 ± 4.7	56 ± 5.1	76 ± 6.7	32 ± 1.7
40	1	White	985 ± 0.03	11 ± 0.10	3097 ± 607.2	3797 ± 1549.2	1039 ± 159.9	339 ± 181.0	1478 ± 331.8	3 ± 1.3	45 ± 29.0	27 ± 4.6	54 ± 14.6	23 ± 4.7
	2		991 ± 0.44	16 ± 0.50	1623 ± 10.0	1935 ± 0.7	897 ± 0.1	169 ± 6.5	836 ± 0.3	2 ± 0.0	29 ± 0.1	8 ± 0.3	39 ± 0.1	8 ± 1.1
	3		991 ± 0.22	17 ± 0.08	8420 ± 1164.2	8523 ± 118.2	1352 ± 74.4	626 ± 17.3	3534 ± 205.8	13 ± 1.2	101 ± 26.0	54 ± 2.3	71 ± 4.0	31 ± 1.8
41	1	White	985 ± 0.03	18 ± 0.01	2826 ± 573.1	3494 ± 1644.0	1095 ± 175.4	372 ± 34.6	1419 ± 199.1	3 ± 1.0	36 ± 8.8	16 ± 4.1	44 ± 27.3	14 ± 3.3
	2		990 ± 0.05	10 ± 0.12	1780 ± 501.6	1822 ± 434.0	1046 ± 84.1	180 ± 125.6	922 ± 282.4	3 ± 0.4	29 ± 6.8	10 ± 2.6	24 ± 1.5	8 ± 2.5
	3		991 ± 0.28	13 ± 0.23	7162 ± 106.8	8171 ± 47.7	1233 ± 40.7	678 ± 195.6	3432 ± 73.5	8 ± 0.1	100 ± 0.1	49 ± 9.5	36 ± 3.6	35 ± 1.6
42	1	White	986 ± 0.02	15 ± 0.04	3324 ± 1692.0	5285 ± 2440.3	1269 ± 256.2	605 ± 254.2	1188 ± 125.8	4 ± 2.2	45 ± 7.4	40 ± 31.4	53 ± 18.0	13 ± 1.8
	2		99.1 ± 0.02	16 ± 0.32	2492 ± 861.3	2522 ± 704.7	1037 ± 73.4	365 ± 132.1	968 ± 308.0	2 ± 0.7	29 ± 6.8	14 ± 48.2	51 ± 9.1	8 ± 2.7
	3		990 ± 0.82	20 ± 0.68	7162 ± 870.8	8003 ± 195.9	1224 ± 146.2	641 ± 127.0	3379 ± 191.8	6 ± 0.3	112 ± 4.2	48 ± 4.0	76 ± 10.3	35 ± 2.9
43	1	White	985 ± 0.01	10 ± 0.01	3657 ± 1946.0	5879 ± 2889.8	1188 ± 256.3	555 ± 291.0	1948 ± 808.9	6 ± 2.6	34 ± 10.5	24 ± 8.1	58 ± 14.7	23 ± 8.5
	2		991 ± 0.01	10 ± 0.44	1878 ± 501.2	2501 ± 496.8	1037 ± 73.4	480 ± 83.2	1035 ± 275.4	3 ± 0.6	29 ± 6.5	12 ± 3.7	53 ± 6.3	7 ± 1.9
	3		990 ± 0.06	15 ± 0.23	10979 ± 1088.0	11084 ± 622.9	1643 ± 115.1	567 ± 101.0	4099 ± 138.9	13 ± 2.1	76 ± 4.7	78 ± 3.8	79 ± 8.9	44 ± 0.3
44	1	White	985 ± 0.01	10 ± 0.08	3699 ± 1957.2	2632 ± 497.4	1209 ± 222.1	500 ± 170.5	1987 ± 779.3	4 ± 1.8	39 ± 15.1	25 ± 6.6	59 ± 17.0	16 ± 4.6
	2		990 ± 0.00	8.8 ± 0.48	2194 ± 550.9	2491 ± 529.5	1161 ± 59.9	338 ± 48.3	1062 ± 273.4	3 ± 0.6	32 ± 7.2	15 ± 2.4	46 ± 5.0	9 ± 1.5
	3		990 ± 0.01	24 ± 0.85	6012 ± 241.1	9196 ± 645.7	1506 ± 122.4	958 ± 83.7	2936 ± 144.9	9 ± 0.5	77 ± 5.6	41 ± 2.5	87 ± 9.9	25 ± 2.3
45	1	White	984 ± 0.07	10 ± 0.06	2384 ± 280.0	2968 ± 815.4	1086 ± 23.6	449 ± 121.8	1458 ± 313.6	3 ± 0.7	34 ± 7.8	17 ± 3.6	58 ± 14.1	19 ± 1.0
	2		991 ± 0.03	9.2 ± 0.36	1320 ± 432.4	1767 ± 487.3	933 ± 99.2	405 ± 21.6	729 ± 289.0	2 ± 0.5	25 ± 7.3	11 ± 2.8	63 ± 0.9	6 ± 2.0
	3		991 ± 0.26	18 ± 0.64	6829 ± 1079.8	9047 ± 403.7	1449 ± 56.2	853 ± 32.0	2904 ± 116.6	9 ± 0.4	100 ± 11.4	49 ± 0.6	74 ± 0.8	28 ± 1.0
46	1	White	985 ± 0.10	18 ± 0.03	2532 ± 271	4341 ± 1161.4	1121 ± 260.9	513 ± 97.2	2117 ± 918.6	4 ± 2.2	40 ± 14.5	20 ± 1.7	75 ± 9.7	17 ± 2.6
	2		990 ± 0.04	7 ± 0.35	1380 ± 359.8	1620 ± 317.6	849 ± 33.7	410 ± 43.7	772 ± 214.0	3 ± 0.3	23 ± 4.2	9 ± 2.5	68 ± 4.6	5 ± 1.4
	3		990 ± 0.88	14 ± 0.09	6737 ± 272.6	9543 ± 628.0	1567 ± 31.6	747 ± 83.2	3102 ± 128.5	7 ± 0.6	79 ± 4.8	50 ± 2.6	88 ± 4.5	26 ± 0.9
47	1	White	986 ± 0.63	17 ± 0.04	1801 ± 641.1	3155 ± 297.8	1056 ± 325.2	508 ± 126.7	1852 ± 853.4	4 ± 1.7	39 ± 5.3	21 ± 8.0	67 ± 10.9	16 ± 3.4
	2		981 ± 0.11	13 ± 0.12	1787 ± 433.6	2731 ± 595.1	849 ± 62.1	480 ± 14.3	940 ± 235.8	3 ± 0.1	28 ± 5.7	10 ± 2.3	53 ± 4.0	11 ± 2.6
	3		991 ± 0.02	15 ± 0.30	6695 ± 170.0	8038 ± 561.7	1401 ± 23.9	683 ± 65.7	3147 ± 109.1	9 ± 0.4	77 ± 4.2	44 ± 2.8	87 ± 10.3	26 ± 2.3
48	1	White	985 ± 0.30	18 ± 0.52	3090 ± 593.8	3001 ± 686.4	934 ± 115.0	492 ± 17.2	1553 ± 154.5	4 ± 1.0	49 ± 1.4	21 ± 5.8	58 ± 7.0	14 ± 4.0
	2		991 ± 0.05	12 ± 0.28	1360 ± 471.4	1434 ± 508.1	858 ± 22.4	478 ± 83.3	523 ± 303.4	2 ± 0.5	20 ± 7.4	6 ± 2.3	53 ± 5.9	4 ± 2.0
	3		990 ± 0.78	15 ± 0.05	6198 ± 183.4	9407 ± 821.4	1417 ± 14.1	657 ± 65.7	2923 ± 730.0	7 ± 1.3	80 ± 9.7	36 ± 2.1	87 ± 12.1	38 ± 0.8
49	1	White	987 ± 0.04	10 ± 0.36	2898 ± 420.0	2978 ± 343.2	1043 ± 327.0	549 ± 134.3	2047 ± 115.2	3 ± 0.70	40 ± 7.1	20 ± 6.0	62 ± 10.0	19 ± 6.2
	2		991 ± 0.01	13 ± 0.01	1364 ± 507.3	1628 ± 318.8	820 ± 43.3	475 ± 126.4	719 ± 300.4	2 ± 0.5	24 ± 7.5	9 ± 3.7	56 ± 8.4	4 ± 1.5
	3		991 ± 0.10	13 ± 0.10	9904 ± 197.1	9266 ± 915.8	1311 ± 19.3	670 ± 46.9	3395 ± 194.4	10 ± 2.0	98 ± 7.2	59 ± 7.3	77 ± 3.5	29 ± 1.5
50	1	White	988 ± 0.34	16 ± 0.05	2887 ± 0.15	3867 ± 1131.4	954 ± 66.1	635 ± 13.0	1326 ± 16.0	4 ± 0.6	34 ± 0.4	15 ± 1.0	64 ± 5.0	10 ± 0.2
	2		990 ± 0.02	12 ± 0.28	1103 ± 412.2	1482 ± 287.8	734 ± 26.7	400 ± 78.9	531 ± 230.4	2 ± 0.5	17 ± 4.7	5 ± 2.0	55 ± 3.6	5 ± 0.3
	3		991 ± 0.24	15 ± 0.16	7989 ± 770.3	7582 ± 187.9	1506 ± 15.5	652 ± 28.2	3733 ± 220.5	11 ± 3.1	99 ± 5.9	6 ± 0.2	58 ± 5.7	59 ± 2.5
51	1	White	982 ± 0.01	20 ± 0.04	3307 ± 1252.0	3058 ± 638.0	1029 ± 258.0	520 ± 111.0	1907 ± 654.4	3 ± 0.8	40 ± 11.1	24 ± 7.0	68 ± 14.5	15 ± 3.8
	2		991 ± 0.02	9 ± 0.14	1580 ± 265.9	2726 ± 379.1	875 ± 48.7	448 ± 23.1	929 ± 5.5	3 ± 0.1	33 ± 4.5	12 ± 3.9	61 ± 9.0	8 ± 0.8
	3		990 ± 0.30	14 ± 0.63	12,380 ± 227.3	8760 ± 631.8	1519 ± 86.4	773 ± 121.9	653 ± 45.2	8 ± 0.5	58 ± 7.1	29 ± 2.5	63 ± 12.3	34 ± 8.2
Ground grains	-	-	<0.001	<0.001	0.58	0.81	0.32	0.55	0.02	0.04	0.05	0.53	<0.001	0.28
Endosperm	-	-	<0.001	<0.001	0.76	0.08	<0.001	<0.001	0.93	0.54	<0.001	0.88	<0.001	0.04
Bran	-	-	<0.001	0.20	0.69	0.73	0.30	0.55	<0.001	0.05	0.53	0.006	0.04	0.02

Numbers 22–51 were on the R (restoring line), numbers refer to 1: whole grains (ground), 2: endosperm (grits), 3: bran. Values of the mineral mg/kg^−1^ presented as mean ± SD, while the dry matter and protein % were analyzed based on two replication readings, *p* ≤ 0.05.

## Data Availability

The original contributions presented in the study are included in the article, further inquiries can be directed to the corresponding author.

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
