# Peer review of "Revealing Consequences of the Husking Process on Nutritional Profiles of Two Sorghum Races on the Male Sterility Line"

_foods, 2024, doi:10.3390/foods13071100_

Round 1
Reviewer 1 Report
Comments and Suggestions for Authors
Dear authors
I read article with subject Revealing Consequences of the Husking Process on Nutritional Profiles of Two Sorghum Races on the Male Sterility Lin. the article is suitable to publish.
please revised the references based on MDPI journal.
let me know what is novelty of this research?please add in your article.
please separate result and discussion
please reduce conclusion to 2 paragraph maximum
lines 450-562 is not clear please revise.
lines 334-345 Cooper section need to revise based on the new references.
Author Response
|
I value your effort in reviewing this work. Below are the comprehensive replies and the appropriate updates and corrections marked in the re-submitted files using tracking changes. Reviewer 1 |
|
Comments 1: [please revise the references based on the MDPI journal.] |
|
Response 1: Thank you for pointing this out. I/We agree with this comment. Therefore, we have.[changed and updated the reference style, using MDPI style references.] |
|
Comments 2: [let me know what is novelty of this research?please add in your article..] |
|
Response 2: We are grateful for the reviewer's concern about the research's uniqueness. We have addressed the study's novelty in the conclusion section. The sentence is: “[The project investigated a novel dehulling procedure for sorghum grain processing, specifically on sorghum-inbred male sterility line varieties. That aimed to establish a sustainable husking process and contribute to sorghum hybrid development programs by assessing the nutritional properties.]” Comments 3: [please separate result and discussion] Response 3: We appreciate the reviewer's recommendation. We have presented the results and discussion independently. Comments 4: [please reduce conclusion to 2 paragraph maximum] Response 4: We considered the reviewer's recommendation and summarized the conclusion section into 2 paragraphs. Comments 5: [lines 450-562 are not clear. Please revise.] Response 5: We considered the reviewer's inquiry for unclear phrases and revised the sentences. Comments 6: [lines 334-345 Cooper section need to revise based on the new references.] Response 6: We considered the reviewer's recommendation, and the sentences were revised to separate results from discussion. |

Reviewer 2 Report
Comments and Suggestions for Authors
THE WORK HAS SOLIDITY AND INOVATIONS. HOWEVER, IT IS NECESSARY INCLUDE LITTLES CORRECTIONS AND SUGGESTIONS THERE ARE MARKED IN TWO ATTACHED COPIES: COPY 01: GENERAL CONSIDERATIONS AND COPY 02: CONSIDERATIONS RELATED TO STATISTICS.

Author Response
|
|
Thank you very much for taking the time to review this manuscript. Please find the detailed responses below and the revisions and corrections highlighted in track changes in the re-submitted files. I would like to inform the reviewer that the language of the whole article was enhanced, the required paraphrasing was set, and the corrections involved the requested recommendations and answering of the respective questions. |
||
|
|
2. Point-by-point response to Comments and Suggestions for Authors: Reviewer 2: |
||
|
|
Comments 1: [It was also unclear why kafirin races were used. How important is the use of kafirin races in the context of this work? This should be clarified in the introduction and discussion] |
||
|
|
Response 1: Thank you for pointing this out. We agree with this comment and the importance of clarifying the points. Therefore, we have….[We addressed clarifying the Kafirin's importance in the introduction and the conclusion]. “[updated text in the introduction is For example, Kafirins, known as the main storage proteins in sorghum, significantly influence its qualitative properties, such as their appropriateness for food, feedstock, and biomaterial uses, which makes them an essential factor in sorghum quality and improvement measurements [43].]” |
||
|
|
Comments 2: [References and citations are not presented in accordance with the format recommended in the instructions for authors. Please review.] Response 2: In response to the reviewer's recommendation, the references were reformatted as journal-desired references. |
||
|
|
Comments 3: [The presentation of the results could be clearer, and the discussion could be more superficial. I strongly suggest that the authors separate this section; that is, results and the discussion must be presented separately to value the results found and discuss them broadly.] Response 3: We appreciate the valuable recommendation of the reviewer. Accordingly, we separated the results and discussion chapters, which helped present the results properly. |
||
|
|
Comments 4: [References and citations are not presented in accordance with the format recommended in the instructions for authors. Please review] |
||
|
|
Response 4: Thank you for pointing this out. We agree with this comment and the importance of clarifying the points. Therefore, we have [reformatted and set the journal reference style]. |
||
|
|
Comments 5: [The abstract should report better the results obtained, i. e., results should be organized to facilitate their understanding. Please review also.] Response 5: Thank you for pointing this out. We agree with this comment and the importance of clarifying the points. Therefore, we have [revised the abstract section by inserting and presenting appropriate results and making grammar corrections]. |
||
|
|
Comments 6: [Line 12: “sorghum bicolor” – please rephrase to “Sorghum bicolor] |
||
|
|
Response 6: Thank you for pointing this out. Due to grammar mistakes, We rephrased the sentence to “Sorghum bicolor]. |
||
|
|
Comments 7: [The introduction section should be more objective and provide more details on recent studies that demonstrate that the Husking Fraction Time Units technique can improve the nutritional content of grains by promoting dietary properties that contribute to food security and economic development. The authors should clarify this issue to highlight and justify the importance of this study.] |
||
|
|
Response 7: We agreed on the reviewer's recommendation. Therefore, we cited more related articles to the study objective. |
||
|
|
|||
|
Comments 8: [-Line 42: “The mentioned studies [5; 6] highlight ...” – please rephrase [5;6] to [5,6] according to the format recommended in the instructions for authors and review the entire manuscript.] |
|||
|
Response 8: We agreed on the reviewer's recommendation. We have changed the reference style of the entire article.
|
|||
Comments 11: [Tables 1 and 2:
- a) What are the codes presented in the first column of these tables? This should be clear in the table footer.?.].
Response 11: We thank the reviewer for his inquiry. Therefore, we presented the type of male sterility for each code in the table footer.
Comments 12: [Please indicate in the table the significant values for each variable analyzed..].
Response 12: We thank the reviewer for his inquiry. Therefore, we inserted the significance value for each variable in Tables 1 and 2.
Comments 13: [Lines 225-397: I strongly suggest reviewing this entire item. The presentation and discussion of the results is difficult to understand.., and the conclusion is extensive and resembles the discussion of the results. This section should finalize the findings presented and point out perspectives for the advancement of knowledge in the area studied – please review.].
Response 13: We thank the reviewer for his recommendations. Therefore, we revised the entire results and discussion chapters.
|
4. Response to Comments on the Quality of English Language |
|
Point 1: |
|
Response 1: (We prioritise the important matter of English quality. Thus, the corresponding author engaged with a PhD candidate (Saeed Omer) in the Food Sciences sector to improve the language of the paper. As a result, the PhD candidate was included as a co-author.) |
|
5. Additional clarifications |
|
[We hope that the addressed revisions and corrections have enhanced the paper quality and meet the Foods Journal standard. We will be grateful to the reviewer and editor for any questions, observations or recommendations for further enhancements.] |

Reviewer 3 Report
Comments and Suggestions for Authors
This manuscript reports the effect of using the husking process (employing the Husking Fraction Time Units technique) on the nutritional profiles of two Sorghum bicolor races in male sterility lines. The aims of the manuscript are interesting, and the experimental step was apparently well conducted. However, there are some points that need explanation and/or correction. Please see the comments below.
Initially, I highlight that the manuscript needs extensive spelling and grammatical review, that is, extensive editing of English language is required. The manuscript has many confusing and incomprehensible sentences, with inappropriate verb tenses, which made it difficult to read and understand all sections (abstract, introduction, methodology, results/discussion, and conclusion). Some examples can be seen in the lines 15-17, 53-54, 60-61, 84-85, 87-93, 112-115, 176-172, 398-403 …
It was also unclear why kafirin races were used. How important is the use of kafirin races in the context of this work? This should be clarified in the introduction and discussion.
The presentation of the results is confusing, and the discussion is superficial. I strongly suggest that the authors separate this section, that is, results and the discussion must be presented separately to value the results found and discuss them broadly.
References and citations are not presented in accordance with the format recommended in the instructions for authors. Please review.
Please see also:
1. The abstract should report better the results obtained, i. e., results should be organized to facilitate their understanding. Please review also:
- Line 12: “sorghum bicolor” – please rephrase to “Sorghum bicolor”.
- Lines 14-15: “…on the variation in the nutritional profile of fifty-one inbred line sorghum varieties...” - this information is not clear in the methodology – please review.
- Lines 22: S. bicolor – please use italic form (S. bicolor).
- Line 23: S.bicolor – please rephrase to S. bicolor.
2- The introduction section should be more objective and provide more details on recent studies that demonstrate that Husking Fraction Time Units technique can improve the nutritional content of grains by promoting dietary properties that contribute to food security and economic development. The authors should clarify this issue to highlight and justify the importance of this study. Please see also:
- Line 34: “Sorghum bicolor L. ...” – please rephrase to “Sorghum bicolor L.”.
-Line 42: “The mentioned studies [5; 6] highlight ...” – please rephrase [5;6] to [5,6] according to the format recommended in the instructions for authors and review the entire manuscript.
4. The Material and Methods section should be improved, and the questions below should be corrected and/or introduced in this section:
- Please initially provide a subsection on the chemicals used in the different experimental steps, containing information about the manufacturer, city, and country in parentheses.
- Please provide in the text the model, manufacturer, city, and country of the equipment used, in parentheses, the first time they are mentioned.
- Lines 103-106: please rephrase this sentence and provide more detailed information about the samples used in the experimental stage.
- Line 105: Why use kafirin races in late ripping stage?
- Lines 112-115 and 127-129: please rephrase these sentences.
- Line 137: “The grain samples collected during the experiments were analysed at the Central ...” – please rephrase to: “The mineral contentes in the grain samples were analysed at the Central ...”.
Lines 150-151: nitric acid (HNO3, 69% v/v) and H2O2, 30% v/v – please rephrase to nitric acid (HNO3, 69% v/v) and H2O2, 30% v/v.
Line 158: “Evaluation of hulled S.bicolor grain colour” – please rephrase to: “Evaluation of hulled grain colour”.
5. Results and discussion
I suggest that this section be completely rephrased. As mentioned previously, I suggest that the results be presented in one section and the discussion in another section separately. The results are discussed quickly and superficially. Please discuss the results in more detail. Likewise, the presentation of results is confusing and there are incomprehensible sentences. Please see also:
- Lines 176-179: incomprehensible sentence – please rephrase.
- Lines 179-182: incomprehensible sentence – please rephrase.
- Line 184: “...as shown in Figures 1[1,2] and 2 [1,2]” – please avoid using square brackets here; use square brackets for citations only.
- Line 184: Figures 1 and 2 must be presented immediately after being mentioned in the text.
- Tables 1 and 2:
a) What are the codes presented in the first column of these tables? This should be clear in the table footer.
b) Please rephrase mg/kg-1 to mg/kg-1.
c) Please indicate in the table the significant values for each variable analyzed.
d) Please check all results presented in Tables 1 and 2; there are many incomprehensible results such as 4±.23, 23±00.31 and many others.
e) Are the results presented as the mean±standard deviation? This should be clear in the footer.
- Lines 195-197: please rephrase this sentence.
- Line 198: “... the respective varieties, P > 0.05” – please rephrse to: “the respective varieties (P > 0.05).
- Lines 201-205: incomprehensible sentence – please rephrase.
- Lines 207-213: incomprehensible sentence – please rephrase.
- Lines 214-216: incomprehensible sentence – please rephrase.
- Line 217: “showed that Kiffin races have an abundant protein contente...” – please review the term “kiffin races” - wouldn't it be kafirin races?
- Lines 225-397: I strongly suggest that this entire item be reviewed. The presentation and discussion of the results is difficult to understand.
- Lines 398-403: incomprehensible sentence – please rephrase.
6- Conclusion section: the conclusion is extensive and resembles the discussion of the results. This section should finalize the findings presented and point out perspectives for the advancement of knowledge in the area studied – please review.
In my final comments, I highlight that the manuscript needs extensive spelling and grammatical review, that is, extensive editing of English language is required. After this broad review, the manuscript must be resubmitted for further consideration.
Comments on the Quality of English Language
Extensive editing of English language is required.
Author Response
|
Reviewer 3 |
|
|
|
Thank you very much for taking the time to review our manuscript. The author expresses gratitude for the expert review of their manuscript and the valuable comments, stating that the recommendations provided have been thoroughly reviewed and incorporated into both copies. The revised manuscript now aligns better with the standards of (Foods Journal), and the author expresses gratitude for the constructive feedback. We look forward to the opportunity to share our improved manuscript with you in due course. Thank you, |
||

Round 2
Reviewer 3 Report
Comments and Suggestions for Authors
I strongly recommend that authors check in detail all results presented in Tables 1 and 2. I still found many incomprehensible results. As an example, see the Cu column.
Comments on the Quality of English Language
Minor editing of English language required.
Author Response
For research article
|
Response to Reviewer X Comments |
||
|
Reviewer 3 |
|
|
|
We appreciate your effort in reviewing our submissions. The author expresses appreciation for the professional assessment of their paper and the important remarks, which ensured that the document adheres to high standards and is free from errors. Hence, we inform you that we enhanced the revised version based on the suggested corrections, as detailed below: Comment 1: I strongly recommend that authors check in detail all results presented in Tables 1 and 2. I still found many incomprehensible results. As an example, see the Cu column. Response: We express our gratitude to the reviewer for the proposed amendments. In light of these revisions, we have examined the data shown in Tables 1 and 2 and have standardized the decimal format for the mean±SD values according to the respective units of measurement.
Comment 2: Minor editing of English language required. Response: In response for minor editing of English language, we edited and corrected the required English improvements. Thank you, |
||
